# Next Generation Sequencing and Molecular Biomarkers in Ovarian Cancer—An Opportunity for Targeted Therapy

**DOI:** 10.3390/diagnostics12040842

**Published:** 2022-03-29

**Authors:** Laura M. Harbin, Holly H. Gallion, Derek B. Allison, Jill M. Kolesar

**Affiliations:** 1Division of Gynecologic Oncology, Department of Obstetrics and Gynecology, University of Kentucky Markey Cancer Center, 800 Rose Street, Lexington, KY 40536-0596, USA; holly.gallion1@uky.edu (H.H.G.); jill.kolesar@uky.edu (J.M.K.); 2Department of Pathology & Laboratory Medicine, University of Kentucky College of Medicine, Lexington, KY 40536-0596, USA; derek.allison@uky.edu; 3Department of Pharmacy Practice and Science, University of Kentucky College of Pharmacy, 760 Press Avenue, Lexington, KY 40536-0596, USA

**Keywords:** next generation sequencing, precision medicine, ovarian cancer, targeted therapy, PARP inhibitors, immunotherapy, NTRK inhibitors

## Abstract

Ovarian cancer is the deadliest of all gynecologic malignancies claiming the lives of nearly 14,000 women in the United States annually. Despite therapeutic advances, the ovarian cancer mortality rate has remained stagnant since the 1980’s. The molecular heterogeneity of ovarian cancers suggest they may be more effectively treated via precision medicine. Current guidelines recommend germline and somatic testing for all new epithelial ovarian cancer diagnoses to assist providers in identifying candidates for targeted therapies. Next generation sequencing (NGS) identifies targetable, driver, and novel mutations used to guide treatment decisions. Performing NGS is standard of care in many other malignancies, but for ovarian cancer the use of NGS in daily practice is still emerging. This review discusses the targetable genetic mutations and role of NGS and molecular biomarker testing in the treatment of ovarian cancer.

## 1. Introduction

In 2021, approximately 21,000 women in the United States received the diagnosis of ovarian cancer, and nearly 14,000 succumbed to their disease [1]. Ovarian cancer broadly describes a collection of pathologically distinct malignancies. The most common and unfortunately, most deadly type of ovarian cancer is the epithelial-derived high grade serous ovarian carcinoma (HGSOC) [2]. Despite our advances in treatment, we still lack effective screening for ovarian cancer, which results in most patients receiving their diagnosis at advanced stage. Unfortunately, risk of recurrence and death is directly correlated with stage of disease at the time of diagnosis. Patients with early-stage disease have markedly better 5-year survival rates of 90% and 70% for stage I and stage II, respectively. However, over 80% of ovarian cancer patients are diagnosed with advanced stage disease with a 5-year survival rate of 35% or worse. In recent years, the prognosis has improved for many other solid tumors; however, the mortality associated with ovarian cancer has remained stagnant since the 1980’s [3]. Despite the surgical and chemotherapeutic advances of recent years, we still struggle to effectively treat and cure ovarian cancer. It is well documented that ovarian cancer is a molecularly heterogeneous group of malignancies and may be more effectively treated with a precision medicine approach [4].

The advent of Next Generation Sequencing (NGS) has opened the doors to better understand the genetic landscape of malignancies [5]. NGS is a comprehensive and unbiased profile of the cancer genome. It uses massive parallel sequencing to analyze numerous genes simultaneously in a single assay. Due to the limitations of prior sequencing assays, many clinical sequencing options offered single gene or few gene coverage. In comparison, NGS offers an affordable, high throughput, high resolution, and more comprehensive way to sequence large panels of genes and whole exomes [6,7]. NGS can identify driver, targetable, and novel mutations used to guide treatment decisions [6]. Performing NGS is commonplace and is the standard of care in many other malignancies, but for some cancers, including ovarian cancer, the use of NGS in daily practice is still emerging [8]. The most recent NCCN guidelines recommend somatic testing in the up-front setting for *BRCA1*/2 mutations, *NTRK* fusions, homologous recombination deficiency (HRD) and tumor biomarkers including microsatellite instability (MSI), mismatch repair deficiency (MMR) and tumor mutation burden (TMB) for all patients [9]. Current commercially available NGS options often combine whole exome sequencing with immunohistochemical (IHC) testing to report on all available molecular alterations.

The purpose of this article is to review the known targetable genetic mutations in ovarian cancer and discuss the role of NGS and molecular biomarker testing in routine practice when caring for patients with ovarian cancer.

## 2. Next Generation Sequencing

### Testing Modalities

The term “next generation sequencing” recognizes the progression in the last decade from single gene sequencing to high throughput, unbiased, parallel sequencing of entire cancer genomes and transcriptomes. NGS has helped to identify genomic signatures and driver mutations for various cancers. Table 1 summarizes the most common genetic and molecular alterations found in ovarian cancer. NGS reveals these molecular targets and provides clinicians with options for targeted therapy. There are several commercially available NGS testing platforms utilizing different sequencing technologies. The table below summarizes some of the most common clinical NGS testing options and their characteristics (Table 2).

Providers have the option of submitting formalin fixed paraffin embedded tissue (FFPE) or a peripheral whole blood sample for a “liquid biopsy”. In general, FFPE tissue samples grant a more comprehensive gene assessment than liquid biopsies. The primary difference among the three FFPE testing platforms listed is the number of genes assessed and current FDA approval.

Currently, FoundationOne^®^ Companion Diagnostic (CDx) has full FDA approval as a companion diagnostic for all solid tumors. The test identifies over 20 FDA approved targeted therapies based on patient specific genomic signatures in 324 assessed genes. However, the lack of whole transcriptome sequencing may lead to inadequate coverage to detect various gene fusions. CARIS Life Sciences^®^ received device breakthrough designation for MI Transcriptome™ CDx in March 2020 and is awaiting full FDA approval for MI Exome™ CDx and MI Transcriptome™ CDx. Their MI Profile includes whole exome sequencing, whole transcriptome sequencing, and targeted IHC able to detect genetic alterations in 592 genes. Similarly, Tempus awaits full FDA approval of their 648 NGS gene panel, the xT-Onco Assay, which also includes whole exome, whole transcriptome, and targeted IHC testing. All FFPE based NGS tests include or offer genomic signature testing for homologous recombination deficiency (HRD), microsatellite instability (MSI), tumor mutation burden (TMB), and programmed death ligand 1 (PD-L1). For ovarian cancer patients, the CARIS MI Profile™ reflexively includes mismatch repair (MMR), estrogen receptor (ER), and progesterone receptor (PR) testing.

The liquid diagnostic tests, FoundationOne^®^ Liquid CDx, Guardant360^®^, and Tempus xF, offer fewer gene assessments. But, for patients without a tissue specimen they offer a reasonable alternative for providers. Testing is performed on circulating cell-free DNA (cfDNA) isolated from peripheral whole blood of cancer patients. When assessing tumor mutation burden -high (TMB-H) in liquid biopsy samples, samples with at least 16 mut/Mb are considered TMB-H as opposed to solid tumor samples where greater than 10 mut/Mb constitutes TMB-H. As providers select the best testing modality for their practice, it is important to consider the number of genes and biomarkers assessed, but also the sensitivity of the selected test. Studies suggest that the sensitivity to detect microsatellite instability-high (MSI-H) tumors with Guardant360^®^ is 87% compared to Tempus xF with sensitivity of 37.5% for MSI-H [10,11]. Ultimately providers will have to decide which platform offers the best supplement to their practice.

## 3. Genomic Alterations and Targetable Therapeutics in Ovarian Cancer

### 3.1. Molecular Pathogenesis of Ovarian Cancers

The advent of precision medicine with somatic tumor testing and genomic sequencing has helped elucidate the molecular pathogenesis of ovarian cancers [12]. Epithelial ovarian malignancies are classified as type I or type II tumors based on the histology, grade, and molecular alterations present [12]. It was previously taught that type II, poorly differentiated epithelial tumors originated from type I, well differentiated epithelial tumors. It is now accepted that low grade or type I tumors and high grade or type II tumors are molecularly, histologically, and clinically distinct entities [3,13].

Type II tumors, including high grade serous carcinoma, high grade endometrioid, carcinosarcoma, and undifferentiated cancers, typically present at advanced stage and are associated with poor prognosis. Type II tumors frequently demonstrate genomic instability, chromosomal aneuploidy, and *TP53* mutations [13]. Additionally, HGSOC is associated with mutations in *BRCA1/BRCA2*, copy number alterations in cell cycle regulating genes, and deficiency of genes associated with homologous recombination repair (HRR) [3].

In contrast, type I tumors, including low grade serous carcinoma (LGSOC), low grade endometrioid, and mucinous carcinomas often present at early stages and are associated with a more favorable prognosis despite their relative chemo-resistance [13]. The development of LGSOC is associated with mutations in well-known oncogenes and activation of the MAPK pathway [14]. Cheasley et al. assessed the genomic alterations in 71 patients with LGSOC and found activating mutations of the *RAS/RAF* pathway genes in 47% of their population [15]. The indolent nature of LGSOC and associated resistance to chemotherapy makes these tumors prime targets for genomic sequencing and targeted therapeutics in the future [14].

The molecular pathogenesis of rare ovarian malignancies is less well defined. However, some have associated pathognomonic genomic alterations. Nearly all adult type granulosa cell tumors contain a missense mutation of the *FOXL2* gene, which encodes a transcription factor essential for granulosa cells [13]. Approximately 60% of sertoli-leydig cell tumors possess *DICER1* mutations [13]. Widespread use of genomic sequencing will continue to clarify and identify driver mutations in ovarian carcinomas.

### 3.2. Homologous Recombination Repair, PARP Inhibitors, and Role of BRCA1/2

#### 3.2.1. Homologous Recombination Repair and Deficiency

Cell cycle checkpoints allow cells to repair damaged DNA or induce cell death if repair is not possible. Functional DNA repair is paramount for normal cells, and dysfunction can lead to malignant transformation. There are five pathways for DNA repair including mismatch repair (MMR), nucleotide excision repair (NER), base excision repair (BER), non-homologous end joining (NHEJ), and homologous recombination repair (HRR) (Figure 1 and Figure 2) [13,16]. HRR repairs double strand DNA (dsDNA) breaks in the S and G2 phase checkpoint using the sister chromatid as a template (Figure 1), resulting in error free DNA repair [16]. Following recognition of a dsDNA break, the cell recruits several proteins involved in HRR including BRCA1, BRCA2, and PARP1 [16]. Failure of HRR can occur from loss of function mutations in *BRCA1* or *BRCA2*, and other moderate penetrance genes including *RAD51C, RAD51D*, and *PALB2* [17,18]. These mutations confer a “BRCAness” phenotype making these tumors more susceptible to platinum based chemotherapy and responsive to PARP inhibition [17,19,20]. Additional biomarkers related to HRD include genomic instability scores (GIS) and loss of heterozygosity scores (LOH). These scores represent the percentage of a tumor genome with focal loss of a single gene allele. These losses lead to “genomic scars” that occur when cells are HRD positive and unable to repair double-strand DNA breaks [19,20]. Identifying homologous recombination deficient tumors can be done by three methods: HRR gene level testing, “genomic scar” and signature testing, and functional assays [17,19]. Nearly all NGS testing options combine these testing methods into composite tests able to detect HRD gene deficiency, GIS and/or LOH [17]. Although individual tests for HRR gene levels, copy number based “scar” assays, and functional assays exist, the efficiency of comprehensive HRD testing with NGS is unparalleled.

#### 3.2.2. HRD Incidence in HGSOC

The Cancer Genome Atlas (TCGA) estimates that 50% of HGSOC contain abnormalities in the HRD pathway [16,17,21]. Germline mutations in *BRCA1* and *BRCA2* represent 12–15% of this population and somatic *BRCA* mutations identify another 5–7% of cases [17]. The remaining 20–30% of patients exhibit HRD through mutation or silencing of another homologous recombination gene [17,21]. Ledermann et al. identified 12 specific non-*BRCA* HRR genes in epithelial ovarian cancer. Although the observed frequencies range from 0.5–2%, taken together they represent a large portion of HRD tumors, and can potentially identify numerous patients that could benefit from targeted therapy (Table 3) [16,19]. 

#### 3.2.3. PARP Inhibitors Targeted Therapy for HRD

PARP inhibitors (PARPi), initially developed as chemo-sensitizers, were found to demonstrate activity in *BRCA* mutated cells in 2005. Bryant et al. found that *BRCA1/2* deficient cells were 100- to 1000-fold more sensitive to PARP inhibition than *BRCA* wild- type (*BRCA*-WT) cell lines [22,23]. PARP inhibitors exhibit anti-tumor activity by a process known as “synthetic lethality”—the idea that cancer cell inactivation of two genetic pathways leads to cell death while disruption of one pathway alone is non-lethal (Figure 3) [16,23]. PARP enzymes are responsible for base excision repair in single strand DNA (ssDNA) breaks (Figure 3) [16]. When left un-repaired, ssDNA breaks become dsDNA breaks which require repair via homologous recombination repair or non-homologous end joining. In *BRCA* mutated or HRD tumor cells this pathway is disrupted as well leading to the accumulation of many dsDNA breaks [16]. The natural response of cells with accumulated DNA damage is to activate the intrinsic apoptosis pathway leading to cell death. Together, an HRD mutation and PARP1 inhibition is lethal to cancer cells [13,16].

There are currently three PARP inhibitors with FDA approval for use in ovarian cancer: olaparib, niraparib, and rucaparib. Although they have similar anti-tumor effects, they have specific indications for use from maintenance therapy to treatment of recurrent disease.

The ARIEL2 trial, a phase II, open label study, evaluated the efficacy of rucaparib in recurrent, platinum-sensitive HGSOC. This trial categorized patients into one of three HRD groups: *BRCA* mutated, *BRCA* WT/LOH high, and *BRCA* WT/LOH low [21]. *BRCA* mutated subjects experienced the greatest benefit in progression free survival (PFS) at 12.8 months (hazard ratio [HR] 0.27, 95% confidence interval [CI] 0.16 to 0.44, *p* < 0.0001). *BRCA* WT/LOH high subjects also had a statistically significant improvement in PFS at 5.7 months (HR 0.62, 95% CI 0.42 to 0.90, *p* = 0.011) compared to the *BRCA* WT/LOH low subgroup (PFS 5.2 months) [21]. ARIEL2 was the first study to demonstrate a PFS benefit for the HRD population by using LOH as a marker for HRD status [21]. They established a LOH score of 16% or greater to define the “LOH high” subgroup and the patients most likely to receive the PFS benefit from PARPi [21].

Niraparib has also demonstrated improved PFS in the HRD population. The NOVA study, a phase III randomized control trial, compared niraparib to placebo for maintenance therapy in recurrent ovarian cancer. Subjects with germline *BRCA* mutations had the greatest PFS benefit (21.0 months vs. 5.5 months, HR 0.27, 95% CI 0.17 to 0.41, *p* < 0.001). *BRCA* WT, HRD positive patients also experienced improved PFS compared to placebo (12.9 months vs. 3.8 months, HR 0.38, 95% CI 0.24 to 0.59, *p* < 0.001). Even *BRCA* WT, HRD negative patients demonstrated a statistically significant improvement in PFS with niraparib (PFS 9.3 mo vs. 3.9 mo, HR 0.45, 95% CI 0.34 to 0.61, *p* < 0.001) [24]. This study led to FDA approval for maintenance therapy with niraparib in patients with recurrent ovarian cancer regardless of *BRCA* or HRD status.

Currently, nearly all patients diagnosed with HGSOC are candidates for PARPi at some point during treatment. Patients with germline or somatic *BRCA* mutations continue to experience the greatest PFS benefit followed closely by HDR positive and LOH high patients [25]. As a result, it is crucial to identify these patients for targeted therapy with PARPi. Based on the NGS test selected, providers can simultaneously receive information on somatic mutations, HRD status, and LOH status, as well as additional genomic markers and germline mutations.

### 3.3. Microsatellite Instability (MSI) and Mismatch Repair (MMR) as Indications for Immunotherapy

#### 3.3.1. MSI and MMR Function

Human cells are prone to genetic mutations with thousands of mutations estimated to occur daily [13]. In normal cells, mutations are quickly corrected by a sophisticated DNA repair system including mismatch repair (MMR) [13,26]. The primary genes involved in this process include *MLH1, MSH2, MSH6,* and *PMS2* which can be inactivated by germline, somatic, or epigenetic changes [13,26]. Given the multiple pathways to gene inactivation, MMR deficiency is often assessed by protein expression via immunohistochemistry (IHC). However, genomic approaches are increasing in use and are based on the understanding that tumors deficient in MMR accumulate genetic mutations, especially in repetitive DNA sequences known as microsatellites [26,27]. Typically, fives sites in repetitive sequences are analyzed. When instability in two of the five sites is noted, the cancer is considered MSI-High (MSI-H) [13]. This “mutator phenotype” leads to the accumulation of genetic mutations and accelerates the development of malignancies.

#### 3.3.2. MSI-MMR Incidence

MMR deficiency (dMMR) is commonly associated with Lynch Syndrome, a hereditary genetic syndrome defined by a germline mutation in one of the four MMR genes or a heritable deletion of the *EPCM* gene causing *MSH2* silencing [13,28]. Lynch Syndrome has a strong association with colorectal cancer and endometrial cancer with an estimated lifetime risk of 40–60% for each [13]. Women with Lynch Syndrome also have increased lifetime risk of developing ovarian cancer (5–10%), stomach cancer, small bowel cancer, and renal cancer among others [13,28]. However, the majority of tumors with MMR deficiency occur sporadically [26,29]. MMR deficiency is found in 2–4% of all malignancies, which includes both germline and sporadic cases. The incidence and role of dMMR in ovarian cancer has been less studied compared to other malignancies. Studies report an incidence between 2–29%; however, this wide range is likely due to methodological variability, sample sizes, frequency of different histologic subtypes, and cancer stage [28,29,30,31,32].

While serous ovarian cancer is the most common histology, numerous studies have demonstrated increased incidence of dMMR and MSI-H phenotype in non-serous ovarian cancer [28,29,30,31,32,33,34,35]. A study by Leskela et al. reported the incidence of dMMR based on tumor histology as follows: 18% of endometrioid tumors and 2% of clear cell tumors were dMMR with an overall incidence of 7.5% in their sample population [30]. Xiao et al. published a similar study with 6.9% of all tumors having dMMR. In their sample population, 4.7% of HGSOCs exhibited dMMR compared to 13.3% of endometrioid tumors [34] In addition, they demonstrated higher incidence of dMMR among all non-serous histologic subtypes compared to serous tumors. [34] Given that non-serous tumors are more likely to present at earlier stages, both studies found that dMMR tumors were diagnosed at early stage (I or II) compared to non-serous tumors [30,34].

#### 3.3.3. Use of Immunotherapy in dMMR/MSI-H Ovarian Cancer

Several immunotherapies have been developed to date. While none are specifically approved in ovarian cancer, pembrolizumab has disease agnostic indications and can be used to treat a subset of ovarian cancers. Pembrolizumab is a humanized immunoglobulin G4 monoclonal antibody that binds to programmed death receptor-1 (PD-1) on CD8+ T cells preventing interaction with programmed death ligand-1 (PD-L1) and PD-L2 on tumor cells. This allows re-activation of T-cell mediated tumor destruction [26,36]. The efficacy of pembrolizumab in MSI-H/dMMR tumors has been proven across a wide range of malignancies. The KEYNOTE-158 study enrolled 233 patients with a MSI-H phenotype spanning 27 different tumor types. The objective response rate (ORR) was 34.3% (95% CI 28.3% to 40.8%) [26]. Ovarian cancer patients made up 6% of the sample population and they demonstrated comparable ORR at 33.3% (95% CI 11.8% to 66.5%) [26]. Following this study, pembrolizumab received the first FDA approval for a tumor-agnostic, histology-independent cancer therapeutic in 2017 [26,37].

Dostarlimab-gxly was recently FDA approved for dMMR recurrent or advanced endometrial cancer and other solid tumors. Approval was based on the GARNET study which enrolled 209 patients with dMMR recurrent or advanced solid tumors (103 endometrial and 106 other). The ORR in the non-endometrial cohort was 38.7% (95% CI 29.4 to 48.6 months). The two ovarian cancer patients in this cohort experienced partial response and stable disease respectively [38]. Like pembrolizumab, dostarlimab-gxly is not approved specifically for ovarian cancer, but it has disease agnostic approval for dMMR tumors. Additionally, phase 1 studies have demonstrated effectiveness in ovarian cancer, and dostarlimab is listed by the NCCN as a therapeutic option for advanced or recurrent ovarian tumors [39].

### 3.4. Tumor Mutation Burden (TMB) as Biomarker for Immunotherapy

#### 3.4.1. Tumor Mutation Burden as Biomarker

Tumor mutation burden (TMB) is generally defined as the total number of mutations present in a tumor specimen. TMB varies with both the type of sequencing performed and the method used for calculation [40]. TMB can be evaluated by whole exome sequencing which assesses all non-synonymous mutations in coding regions (excluding germline mutations). The results are compared to a matched normal sample, but this method is currently used primarily in a research setting. TMB can also be evaluated by commercial, clinically used NGS panels which targets pre-specified hot spot genes associated with cancer. TMB assessed in NGS panels includes synonymous variants and short indels in intronic regions which are not covered in whole exome tests.

Heavily mutated tumors are known to harbor many neoantigens and upregulate immune checkpoint proteins, resulting in increased T-cell reactivity and correlation with improved responses to immune checkpoint inhibitors (ICI) [40,41,42,43]. While there is variability in TMB assessment and reporting, tumors with at least 10 mutations per megabase (mut/Mb) are considered TMB-high (TMB-H) and have demonstrated greater anti-tumor activity with ICIs [40,43,44]. This correlation and clinical benefit for checkpoint inhibitors in TMB-H tumors has been demonstrated in non-small cell lung cancer and melanoma, among other tumor types, and led to FDA approval of immune-oncologic therapy as first- line treatment in some cases [40,44,45].

#### 3.4.2. TMB-H Incidence

A study by Chalmers et al. found a median TMB in ovarian cancer to be 3.6 mut/Mb; however, this number varies greatly based on the histologic subtype [42]. Contos et al. found that approximately 10% of their patients met criteria for TMB-H with ≥10 mut/Mb. Endometrioid and adenocarcinoma not otherwise specified (NOS) subtypes demonstrated the highest percentage of TMB-H tumors at 15.4% [45]. In addition, 7.9% of clear cell cancers and 4.1% of serous tumors were classified as TMB-H [45]. High TMB has been associated with improved progression free survival (PFS) and overall survival (OS) in other malignancies when treated with ICIs [46,47]. Although TMB has not been exclusively studied in ovarian cancer, identification of TMB-H tumors helps to identify a percentage of patients that could benefit from immunotherapy. 

#### 3.4.3. Use of Immunotherapy in TMB-H Ovarian Cancer

The use of immunotherapy in TMB-H ovarian cancer has not been specifically studied to date. However, a cohort in the KEYNOTE-158 study evaluated the efficacy of pembrolizumab among various solid malignancies. There were 805 patients evaluable for TMB with 105 (13%) categorized as TMB-H with ≥10 mut/Mb. Objective responses were observed in 30 (29%, 95% CI 21% to 39%) of 102 subjects in the TMB-H group compared to 43 (6%, 95% CI 5% to 8%) of 688 subjects in the TMB-low group [40]. However, median PFS was similar between TMB-H and TMB-low groups at 2.1 months (95% CI 2.1 to 4.1 months) and 2.1 months (95% CI 2.1 to 2.2), respectively [40]. Although there were no patients with ovarian cancer in this study, pembrolizumab is still FDA approved for all patients with solid malignancies and TMB ≥ 10 mut/Mb.

### 3.5. Programmed Death Ligand 1 (PD-L1) Expression as Biomarker for Immunotherapy

#### 3.5.1. PD-L1 Expression and Function

PD-1 is a transmembrane receptor protein expressed on the surface of T cells, B cells, and natural killer (NK) cells [26,36,48]. Activation of PD-1 by its ligands PD-L1 and PD-L2 initiates negative regulation of T cell activation. When PD-1 binds PD-L1, a ligand expressed on many tissues including tumor cells, the T cell blocks cell signaling pathways and decreases cell proliferation and survival [49]. However, during states of chronic antigen exposure (such as viral infections or cancer), T cells maintain high expression of the PD-1 receptor leading to functional exhaustion with an inability to secrete cytokines or kill target cells [13,49]. The tumor microenvironment with upregulation of the PD-1 pathway inhibits native antitumor immune response. Blockade of the PD-1 pathway, via ICI, inhibit this ligand-receptor interaction leading to re-activation of the patient’s immune system to target and kill the malignant cells [49].

#### 3.5.2. PD-L1 Expression Incidence in Ovarian Cancer

PD-L1 tumor expression is assessed by IHC; however, there are multiple antibody clones and scoring methods. The 22C3 clone is reported either as the combined positive score (CPS) or the tumor proportion score (TPS) [50]. The CPS evaluates the number of PD-L1 positive cells (including tumor cells, lymphocytes, and macrophages) relative to all tumor cells [50]. The TPS compares the proportion of PD-L1 positive to PD-L1 negative tumor cells. Whether the TPS or CPS is reported depends on tumor type and companion diagnostic status [50]. For example, in non-small cell lung cancer (NSCLC), a TPS ≥ 1% is indication for pembrolizumab therapy, but in cervical cancer, a CPS ≥ 1% is used [50]. The SP142 clone is used to evaluate tumor cells expressing PD-L1 as a percentage of total cancer cells and immune cells expressing PD-L1 as a percentage of tumor area. With this assay, tumors are reported as PD-L1 positive if either tumor cells or immune cells are greater than 1% [50]. While there is not currently an indication for immunotherapy based on the PD-L1 score for ovarian cancer, Contos et al. evaluated the immune biomarkers of 8809 ovarian cancer patients using the SP142 assay. They found that 7.8% of their patients were PD-L1 positive using a CPS cutoff of 5% instead of ≥1% [45]. Germ cell tumors were most likely to be PD-L1 positive (28.9%) followed by ovarian neuroendocrine tumors (14.3%) and clear cell tumors (12.2%). A substantial cohort of epithelial ovarian cancers also demonstrated PD-L1 positivity with 9.2% of endometrioid and 7.4% of serous ovarian cancers having >5% PD-L1 expression [45].

#### 3.5.3. Use of Immunotherapy in PD-L1 Positive Ovarian Cancer

PD-L1 expression in ovarian cancer is an independent predictor of response to immunotherapy and is associated with improved prognosis in HGSOC [45,51,52]. The efficacy of pembrolizumab has been specifically studied in PD-L1 positive, advanced, recurrent ovarian cancer in the KEYNOTE-100 study [53]. This study enrolled 376 patients and found an ORR of 8% (95% CI 5.4 to 11.2) for all subjects. Importantly, they found an improved ORR, 17.1%, for subjects with CPS ≥ 10 that was not impacted by number of prior regimens, progression free interval, platinum-sensitivity status, or tumor histology [53]. Thus, treatment with pembrolizumab for patients with high CPS scores and few alternative therapy options seems reasonable. Another small trial of platinum-resistant ovarian cancer patients demonstrated an ORR of 15% for patients on nivolumab, an anti- PD-1 treatment [54]. Excitingly, two of the twenty patients in the trial experienced prolonged duration of response of 14 and 17 months respectively [54]. Although the use of immunotherapy and PD-L1 as a biomarker in ovarian cancer remains novel, these early trials demonstrate promise for the future. Currently, there are several clinical trials evaluating the efficacy of combination therapy with immunotherapy, anti-angiogenic therapy, chemotherapy, and PARPi in ovarian cancer. Their results will help determine which patients are the best candidates for immunotherapy treatment.

### 3.6. NTRK Mutations

#### 3.6.1. *NTRK* Gene Function

The neurotrophic tyrosine receptor kinases (*NTRK*) genes, *NTRK1*, *NTRK2*, and *NTRK3*, encode three transmembrane proteins important for the development and function of the nervous system [55,56,57]. Together, these three proteins regulate pain and temperature sensations, control movement, aid in memory, and regulate mood, appetite and body weight [55]. Ligand binding of these proteins causes receptor homodimerization followed by phosphorylation leading to downstream activation of cell-signaling pathways such as mitogen-activated protein kinase (MAPK), phosphatidylinositol-3-kinase (PI3K)/AKT [55]. Activation of these pathways are critical for cell migration, cell differentiation, neuronal function, and cell survival [55,58].

Mutations involving the *NTRK* genes include point mutations, splice variations, copy number alterations, and fusion mutations [56,58,59,60]. The clinical implications of single nucleotide variations or copy number alterations is currently unknown [58]. However, *NTRK1-3* gene fusions are known actionable oncogenic driver mutations [56]. The in-frame fusion occurs between the 3′ sequence of the *NTRK* gene and the 5′ sequences of a wide range of gene fusion partners, some of which have yet to be identified [55,60]. This fusion of *NTRK* genes with various 5′ gene partners results in ligand-independent constitutional activation and overexpression of TRK kinase (Figure 4) [56,58,59,60]. This action and subsequent activation of MAPK and PI3K/AKT leads to cancer cell proliferation and failure of apoptosis [59,60].

#### 3.6.2. *NTRK* Gene Fusion Incidence

Malignancies with *NTRK*-fusions fall into two categories. One group includes very rare tumors often characterized by pathognomonic *NTRK*-fusion. These malignancies are associated with *NTRK* fusions in >90% of cases [61]. Secretory breast cancer, secretory cancer of the salivary glands, congenital mesoblastic nephroma, and infantile fibrosarcomas are characterized by *ETV6-NTRK3* fusion mutations [60,61]. The second group includes many common malignancies where incidence of *NTRK*-fusion mutations is much lower, often less than 1%. The long list of malignancies includes non-small cell lung cancers (NSCLC), colorectal cancer, papillary thyroid cancer, breast cancer, melanoma, and ovarian cancer [59,60,61].

#### 3.6.3. Targetable *NTRK* Mutations

Although the incidence of *NTRK*-fusion mutations in ovarian cancer is low, it is important to identify patients that could benefit from treatment with TRK inhibitors. There are currently two FDA approved, orally available, targeted therapies for patients with *NTRK* fusion mutations, larotrectinib and enetrectinib. Larotrectinib, a first-generation selective TRK inhibitor, received accelerated approval from the FDA in 2018 based on data from 55 patients with *NTRK* fusions [57]. Hong and colleagues provided a combined analysis of 159 patients [62]. Patients had advanced or metastatic disease and had received one prior therapy. Among 153 evaluable subjects, the ORR was 79% (95% CI 72% to 85%). 16% (24/153) achieved a complete response, 63% (97/153) achieved a partial response, 12% (19/153) maintained stable disease and 6% (9/153) had progressive disease [62]. This drug offers a unique targeted therapy as it can be applied to all cancers with proven *NTRK* fusion mutations in a disease agnostic fashion [55]. The anti-tumorigenic effects were rapid and sustained with median time to response of 1.8 months and median duration of response 35.2 months [56].

Enetrectinib, another first-generation TRK inhibitor, acts on tumors with *ROS1, ALK,* and *NTRK* gene rearrangements [55]. Entrectinib activity was evaluated in two phase 1 clinical trials, ALKA-372-001 and STARTRK-1 [63]. Among the 25 evaluable patients with an *NTRK, ROS1*, or *ALK* fusion who received the recommended phase 2 dose and had not received a prior TRK inhibitor, the ORR was 100% (95% CI 60% to 90%) in 3 patients with *NTRK* fusions, 86% (95% CI 60% to 96%) in the 14 patients with *ROS1* rearrangements and 57% (95% CI 25% to 84%) in 7 patients with *ALK* rearrangements [63]. Responses were durable with median duration of response over 10 months [56]. Larotrectinib and enetrectinib not only demonstrate efficacy but also favorable adverse effect profiles. The most common grade 3 treatment-related adverse effects included fatigue, weight gain, and anemia [60]. Less severe adverse effects, including nausea, diarrhea, myalgias, arthralgias, paresthesia, and dizziness, were all reversible with dose modifications [60]. The tolerability and duration of response make TRK inhibitors candidates for long term therapy.

TRK inhibitors such as larotrectinib and enetrectinib are two new additions to our precision medicine toolbox. Although *NTRK* fusions are found in a small subset of ovarian cancer patients, the favorable objective response, disease agnostic indication, and few adverse effects with TRK inhibitors should make them a viable choice for patients with metastatic disease and no standard therapy available. One caveat regarding the use of TRK inhibitors in ovarian cancer is the clinical trials supporting the approval of larotrectinib and entrectinib did not include any patients with ovarian cancer. Therefore, further study in ovarian cancer is warranted [64].

## 4. Discussion

Precision medicine is the next frontier in cancer care. While ovarian cancer has lagged behind other solid tumors in the availability of targeted therapies, there are now multiple treatment options including PARP inhibitors, immunotherapy, and NTRK inhibitors in our therapeutic arsenal. Although there is limited clinical trial data, there are reasons to be optimistic. PARP inhibitors have revolutionized the treatment of the *BRCA* mutated population by significantly improving disease control and survival [21,24,65,66,67]. Studies estimate that 50% of ovarian cancer patients exhibit an HRD phenotype via germline *BRCA* mutations, somatic *BRCA* mutations, or abnormal expression of other moderate penetrance genes [16,17,21]. Each of these subgroups are candidates for PARPi and have demonstrated improved progression free survival with treatment.

Although less studied, the correlation between tumor biomarkers and response to immunotherapy also demonstrates promise in the treatment of ovarian cancer. Endometrioid and clear cell carcinomas have increased incidence of dMMR, MSI-high, and TMB-high tumors that could be effectively targeted with immunotherapy [42,45]. PD-L1 expression is present in approximately 8% of ovarian cancers with nearly 30% of germ cell tumors exhibiting PD-L1 positivity in one study [45]. Most germ cell tumors are diagnosed at early stage and effectively cured with standard of care treatment. However, if recurrent, prognosis is poor, and data is limited to guide treatment decisions. NGS and consideration of targeted therapy given the high immunogenicity of many germ cell tumors could be a reasonable alternative [45]. In addition, while *NTRK* fusions are found in a small proportion of ovarian cancers, identification of patients with an *NTRK* fusion can offer a well- tolerated alternative treatment with entrectinib or larotrectinib.

The advent of NGS testing has facilitated the development of precision medicine. Current guidelines recommend that all patients with ovarian, fallopian tube, or primary peritoneal cancer should have genetic risk evaluation with germline and somatic testing at diagnosis [9]. Medical societies have previously recommended germline genetic testing for all new epithelial ovarian cancer diagnoses; however, the recommendation for widespread somatic testing is new [68]. Somatic tumor molecular analysis can be conducted by NGS for *BRCA* mutations, other somatic mutations, and biomarkers including TMB, HRD, and LOH.

The addition of somatic testing, specifically NGS, now requires providers to understand and incorporate genomic testing into their daily practice. Currently only 30% of women diagnosed with epithelial ovarian cancer undergo any recommended genetic testing [68]. Germline genetic testing can pose challenges to patients and providers as it often requires a physician referral and additional appointment. Somatic testing through NGS often uses previously acquired tissue and can be ordered without an additional appointment. Ordering physicians will need to familiarize themselves with the various testing modalities for NGS and choose the one best suited to their practice and their patient. Collaborative approaches, such as molecular tumor boards, can be helpful for physicians when interpreting results and deciding the best treatment [69,70]. Molecular tumor boards (MTB) use a multi-disciplinary approach to assess patient factors and genomic information to make recommendations for patients not responding to standard- of- care therapy. The MTB often includes medical oncologists, surgical oncologists, pathologists, radiologists, pharmacists, genetic counselors, and basic scientists, each of whom offers their expertise to choose the best therapy or clinical trial [69,70]. Ovarian cancers are often discussed at MTB given their molecular heterogeneity, genomic instability, and propensity to develop platinum resistance. In addition, recent studies have shown improved response rates among patients treated with therapies recommended by MTB [70].

In summary, based on current guidelines, patients diagnosed with ovarian cancer should undergo germline and somatic testing. There are numerous options for molecular tumor testing, but next generation sequencing offers the most complete and efficient genomic evaluation. It is critical to test every patient because over 50% will harbor a targetable mutation. Studies and trials are still needed to validate the efficacy of targeted therapies in ovarian cancer. As we learn more about tumor genetics, discovery of new biomarkers and therapeutics will advance our treatment of ovarian cancer.

## Figures and Tables

**Figure 1 diagnostics-12-00842-f001:**
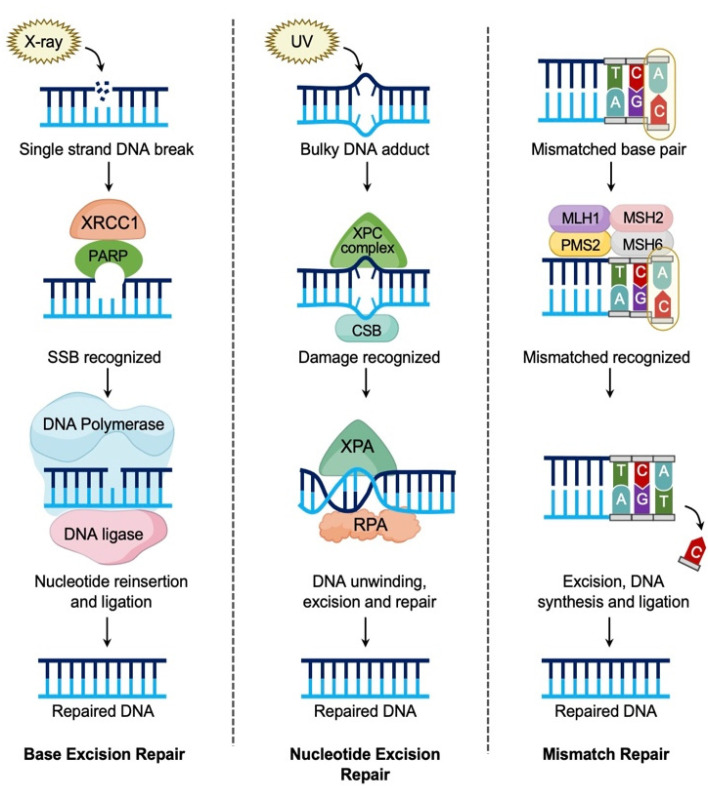
Pathways of DNA repair- single strand DNA breaks. There are three pathways for correction of single strand DNA (ssDNA) breaks. Base excision repair: A single nucleotide base is missing due to spontaneous hydrolysis after DNA damage. XRCC1 and PARP with APE1 endonucleases recognize the damaged site and PARP-mediated repair is initiated. XRCC1 creates a scaffold for DNA polymerase and DNA ligase 3 which reinsert the base and ligate to repair the DNA strand. Nucleotide excision repair: A bulky DNA adduct is generated by environmental carcinogens such as ultraviolet (UV) light. The lesion is detected by xeroderma pigmentosa C complex (XPC complex) and Cockayne syndrome B (CSB) is recruited to the site. The DNA is unwound and XPA and replication protein A (RPA) stabilize the DNA for subsequent proteins to excise, synthesize, and complete the repair. Mismatch Repair: Errors during DNA replication can lead to mismatched base pairs. These mismatches are recognized by MSH2 and MSH6. MLH1 and PMS2 are then recruited to mismatch sites. The incorrect base is excised, replaced, and the corrected strand is ligated.

**Figure 2 diagnostics-12-00842-f002:**
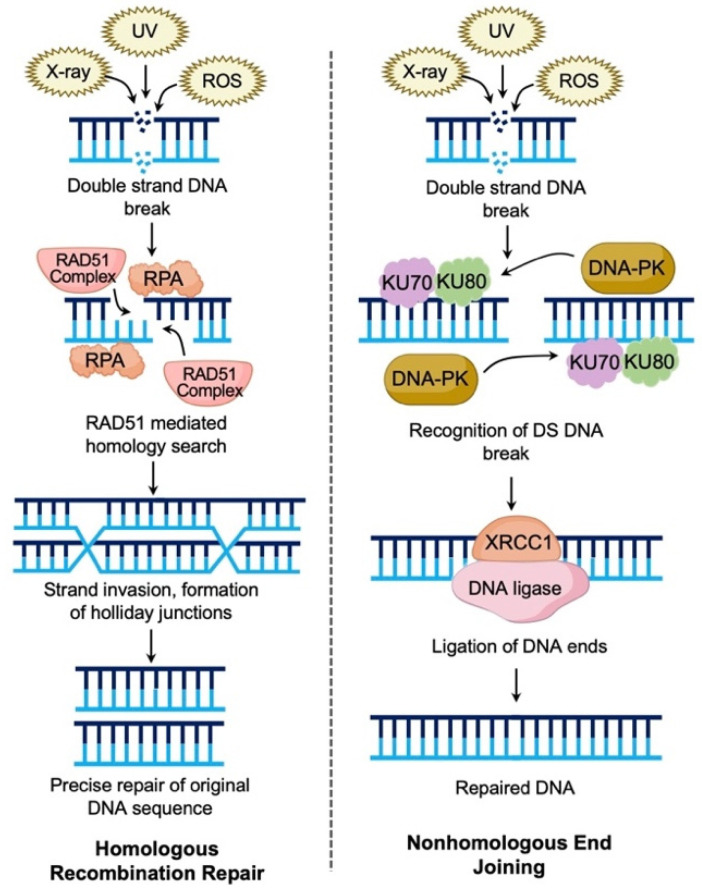
Pathways of DNA repair- double strand DNA breaks. Double strand DNA (dsDNA) breaks can be caused by several exogenous and endogenous factors including radiation, UV rays, chemotherapeutic agents, and reactive oxygen species (ROS). There are two pathways for repair of dsDNA breaks. Homologous recombination repair: The dsDNA break is recognized. Exonuclease activity creates single-strand overhangs that are coated with RPA. The RAD51 complex composed of (RAD52 and BRCA2 proteins) initiate homology search and strand invasion. DNA is synthesized using the sister chromatid as a guide and creating double Holliday junctions. Resolving enzymes correct the junctions creating precisely repaired DNA. Nonhomologous end joining: The dsDNA break is recognized by the Ku-70-Ku80 heterodimer. This recruits DNA protein kinase (DNA-PK). The DNA ends are then ligated together after recruitment of XRCC4 and DNA ligase 4.

**Figure 3 diagnostics-12-00842-f003:**
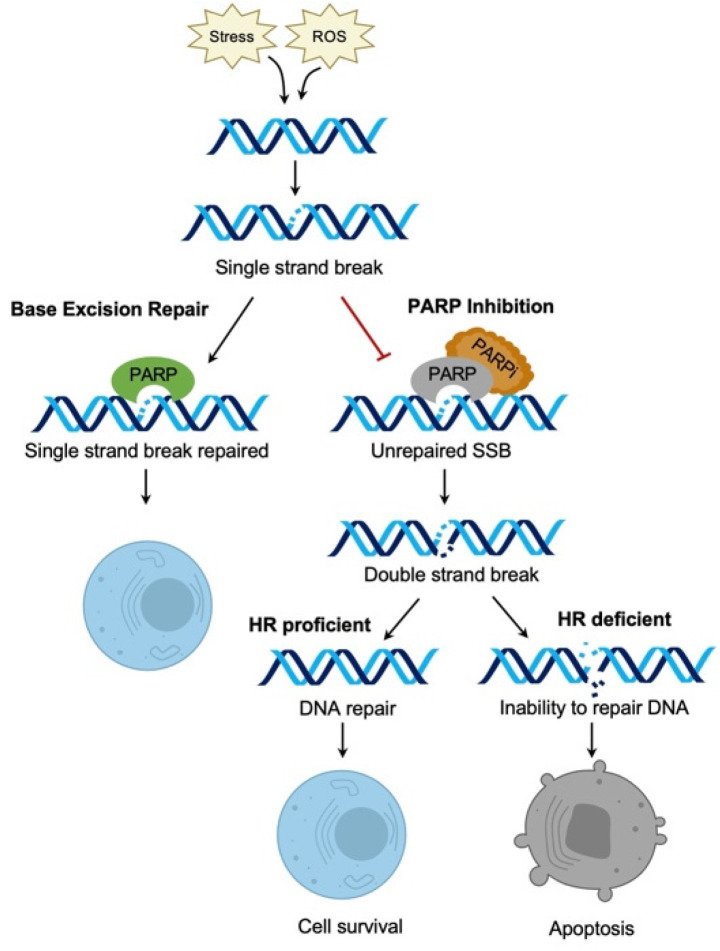
Mechanism of action of PARP inhibitors. PARP proteins are important in the repair of ssDNA breaks. PARP inhibitors trap PARP proteins on ssDNA breaks preventing repair. The ssDNA breaks then become dsDNA breaks. In homologous recombination deficient cells, numerous dsDNA breaks accumulate leading to cell death.

**Figure 4 diagnostics-12-00842-f004:**
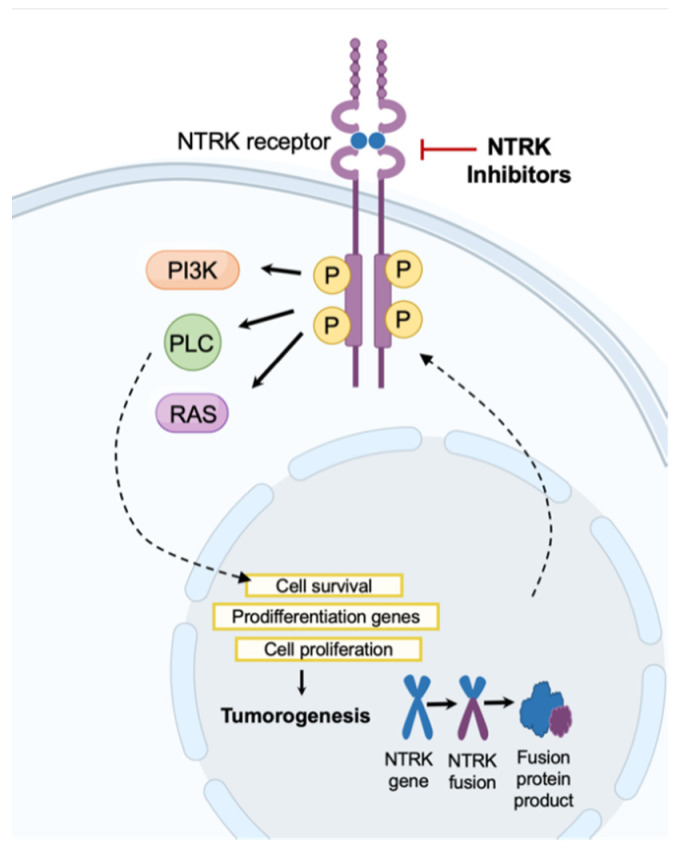
NTRK Inhibitor Mechanism of Action. NTRK inhibitors block action through the NTRK receptor preventing downstream activation of phosphoinositide-3-kinase (PI3K), phospholipase C (PLC), and RAS pathways.

**Table 1 diagnostics-12-00842-t001:** Frequent molecular alterations in the most common histologic subtypes of ovarian cancer.

Histologic Subtype	Frequent Molecular Alterations	Available Targeted Therapies
Epithelial Tumors		
High grade serous carcinoma	*TP53*, *BRCA1*, *BRCA2*, HRR deficiency	PARP inhibitors (1)
Low grade serous carcioma	*KRAS*, *BRAF*, NRAS, *PIK3CA*, *ERBB2*, *PTEN*, *CTNNB1,* ER/PR positive	MEK inhibitors (2) Fulvestrant (2) Hormonal therapy (2)
Clear cell carcinoma	*PIK3CA*, *ARID1A*	None
Endometrioid carcinoma	*CTNNB1*, *ARID1A*, *PIK3CA*	None
Mucinous carcinoma	*KRAS*, *ERBB2*	None
**Germ Cell Tumors**	Karyotypic abnormalities	
Dysgerminoma	*KIT*, *DICER1*, *TP53*, *KRAS*	None
Yolk sac tumor	*KRAS, PIK3CA*	None
**Sex-Cord Stromal Tumors**	*DICER1*	None
Granulosa cell tumors	*FOXL2*, ER/PR positive	Aromatase inhibitors (2) Leuprolide (2)
**Histology Agnostic**		
	TMB-H	Pembrolizumab (1)
	MSI-H dMMR	Pembrolizumab (2) Dostarlimab (2)
	*NTRK* fusions	Larotrectinib (2) Entrectinib (2)

Legend: (1) FDA approved targeted therapy. (2) NCCN guidelines recommended targeted therapy. HRR, homologous recombination repair; ER, estrogen receptor; PR, progesterone receptor; MEK, mitogen activated protein kinase; *NTRK*, neurotrophic tyrosine receptor kinases

**Table 2 diagnostics-12-00842-t002:** Next generation sequencing testing modalities.

Testing Platform	Tissue Type	Genes Assessed	HRD	MSI	TMB	PD-L1	FDA Approval
FoundationOne^®^ CDx	FFPE	324	X	X	X	X ^+^	Yes
CARIS^®^ MI Profile	FFPE	592 *	X	X	X	X	Partial
Tempus xT	FFPE plus blood or saliva	648 **	X ^+^	X	X	X	No
FoundationOne^®^ Liquid CDx	Peripheral whole blood	324		X	X	X ^+^	Yes
Guardant360^®^	Peripheral whole blood	83		X	X		No
Tempus xF	Peripheral whole blood	105		X	X		No

Legend: X: testing included in commercial assay. ^+^: test not included in standard panel but may be added on. *: reflex IHC testing for ovarian cancer patients includes mismatch repair (MMR) testing, estrogen receptor (ER) and progesterone receptor (PR) testing. **: additional IHC options include MMR and PD-L1 testing.

**Table 3 diagnostics-12-00842-t003:** Frequency of genetic mutations in non-*BRCA* HRR genes.

HRR Gene	Mutations per Number of Cases (%)	Number of Unique Mutations	Consequence of Mutation (Frequency/Total # Mut)
*ATM*	15/437 (3.43%)	16	Missense (13/16) Stop gained (1/16) Splice donor deletion (1/16) Intron (1/16)
*BRIP1*	6/436 (1.38%)	6	Missense (4/6) Frameshift (1/6) Intron (1/6)
*CHEK2*	6/436 (1.38%)	7	Missense (4/7) Synonymous (2/7) Splice acceptor (1/7)
*NBN*	4/436 (0.92%)	4	Missense (1/4) Frameshift (1/4) Synonymous (2/4)
*PALB2*	11/437 (2.52%)	12	Missense (7/12) Stop gained (2/12) 3′ UTR (1/12) Splice region substitution (1/12) Protein altering insertion (1/12)
*RAD51B*	3/436 (0.69%)	2	Frameshift (1/2) Intron (1/2)

Legend: 3′ UTR: 3 prime untranslated region. Source: The results published here are in whole or part based upon data generated from The Cancer Genome Atlas (TCGA) Research Network, TCGA-OV project. “https://www.cancer.gov/tcga (accessed on 12 January 2022)”.

## Data Availability

Not applicable.

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
