# Peer review of "Next Generation Sequencing and Molecular Biomarkers in Ovarian Cancer—An Opportunity for Targeted Therapy"

_diagnostics, 2022, doi:10.3390/diagnostics12040842_

Round 1
Reviewer 1 Report
This is an excellent review discussing how targetable genetic mutations and NGS can be used in the personalized treatment of ovarian cancer (OC). The authors have nicely covered broad topics including Homologous Recombination Repair, PARP inhibitors, and BRCA-deficiency, as well as MSI/MMR deficiency as an indication for immunotherapy in ovarian cancer. They also discuss the role of Tumor Mutation Burden (TMB) and PD-L1 expression as biomarkers for immunotherapy in OC. They finally discuss targetable NTRK mutations in the disease.
My main comment for the authors would be to increase the number of papers cited in their bibliography.
Author Response
Reviewer #1
This is an excellent review discussing how targetable genetic mutations and NGS can be used in the personalized treatment of ovarian cancer (OC). The authors have nicely covered broad topics including Homologous Recombination Repair, PARP inhibitors, and BRCA-deficiency, as well as MSI/MMR deficiency as an indication for immunotherapy in ovarian cancer. They also discuss the role of Tumor Mutation Burden (TMB) and PD-L1 expression as biomarkers for immunotherapy in OC. They finally discuss targetable NTRK mutations in the disease.
Author Response: We appreciate the reviewer’s comments and the time to provide us with valuable feedback for our manuscript.
Major comments:
My main comment for the authors would be to increase the number of papers cited in their bibliography.
Author Response: Thank you for this feedback. We have added an additional 14 citations for this review.
Reviewer 2 Report
have read with interest and pleasure this narrative review regarding the application of NGS on ovarian cancer. This neoplasm remains one of the diagnostic and therapeutic challenges of our day, and the authors' treatment was satisfactory with beautiful images and figures that well represent the thread of the discussion. I also really enjoyed the section on immunotherapy and PDL-1, as well as the analysis of NTRK mutations. The part of the discussion takes into consideration the bases that had given support to the data presented and, far from fancy flights, testify the importance of NGS in patients with epithelial ovarian malignancy. I ask the authors to correct some typos and to standardize the references according to the style of MDPI.Author Response
Reviewer #2
I have read with interest and pleasure this narrative review regarding the application of NGS on ovarian cancer. This neoplasm remains one of the diagnostic and therapeutic challenges of our day, and the authors' treatment was satisfactory with beautiful images and figures that well represent the thread of the discussion. I also really enjoyed the section on immunotherapy and PDL-1, as well as the analysis of NTRK mutations. The part of the discussion takes into consideration the bases that had given support to the data presented and, far from fancy flights, testify the importance of NGS in patients with epithelial ovarian malignancy. I ask the authors to correct some typos and to standardize the references according to the style of MDPI.
Author response: We appreciate the reviewer’s comments and the time to provide us with valuable feedback for our manuscript.
We have edited the manuscript to hopefully address all typos. The references have been corrected to the MDPI style as well.
Reviewer 3 Report
The manuscript titled “Next Generation Sequencing for Ovarian Cancer- A review” by Harbin LM et al., is a review article that aim to “discusses the targetable genetic mutations and role of NGS in the treatment of ovarian cancer”. In my opinion the review is incomplete and not clarify the real potential application of NGS.
Major concerns
- Firstly, the manuscript does not clarify the complexity of the different ovarian cancer subtype and the molecular pathogenetic pathway the drive the different subtypes. In fact, beside high grade serous ovarian cancer, low grade serous ovarian cancer and other subtypes shows specific molecular alterations, widely discussed and recognized (as PTEN, PI3K-AKT-mTOR). This point is crucial since other ovarian cancer histotypes and low grade serous are typically chemoresistant and therefore have less therapeutic effective options than high grade ovarian cancer. Therefore, especially for these subtypes NGS could be fundamental to offer personalized targeted approaches and effective treatments in patients that don’t have to date effective therapies.
- Secondly, some of the points discussed are out of the theme of NGS; for example, PDL1 expression is not evaluated by NGS but it is an immunohistochemical assay.
- Third, immunotherapy in ovarian cancer has not demonstrated a role to date. The authors should include a part on the main molecular pathways that drive ovarian cancer carcinogenesis and the specific genetic alteration involved in the two type pathways (high grade and low grade) instead to devote such a large part of the manuscript to immunotherapy
- The paragraph 3 should be moved before after the introduction
- The clinical significance of detect actionable mutations and the incidence of these mutations in ovarian cancer should be discussed. This is fundamental to understand the role of NGS in ovarian cancer, that is not limited only to high grade ovarian cancer and to detect BRCA and HRD mutations.
The review is not acceptable in the present form. It should be completely revised and rewritten
Author Response
Reviewer #3
The manuscript titled “Next Generation Sequencing for Ovarian Cancer- A review” by Harbin LM et al., is a review article that aim to “discusses the targetable genetic mutations and role of NGS in the treatment of ovarian cancer”. In my opinion the review is incomplete and not clarify the real potential application of NGS.
Author response: Thank you for this feedback. I hope that we address your major concerns listed below in a satisfactory manner. Our goal in this review article is to provide practicing clinicians with an accessible guide for targeted therapy in ovarian cancer. The use of NGS in ovarian cancer is still emerging as a tool for clinicians. With the updated NCCN guidelines recommending germline and somatic testing in the front line for all epithelial ovarian cancer patients, we believe this review is timely and needed for the practicing clinician.
Major concerns
Firstly, the manuscript does not clarify the complexity of the different ovarian cancer subtype and the molecular pathogenetic pathway the drive the different subtypes. In fact, beside high grade serous ovarian cancer, low grade serous ovarian cancer and other subtypes shows specific molecular alterations, widely discussed and recognized (as PTEN, PI3K-AKT-mTOR). This point is crucial since other ovarian cancer histotypes and low grade serous are typically chemoresistant and therefore have less therapeutic effective options than high grade ovarian cancer. Therefore, especially for these subtypes NGS could be fundamental to offer personalized targeted approaches and effective treatments in patients that don’t have to date effective therapies.
Author response: Thank you for this feedback. We have added a table to the revised manuscript (table 1) detailing the most common genetic alterations and available targeted therapies based on those alterations. As high grade serous ovarian cancer is the most common and most deadly histologic subtype the majority of research and targeted therapy trials have focused on this group of patients. There have been limited trials evaluating targeted therapies among the other histologic subtypes, but we certainly agree that this chemo-resistant group of patients warrants focus in the future. There are several NCCN guideline recommended therapies for recurrent disease in this subgroup of patients based on molecular alterations in the BRAF or KRAS genes in addition to hormone receptor positivity. We have added these therapies to Table 1.
Secondly, some of the points discussed are out of the theme of NGS; for example, PDL1 expression is not evaluated by NGS but it is an immunohistochemical assay.
Author response: Thank you for this valid point. We have amended out title to read “Next generation sequencing and molecular biomarkers in ovarian cancer- an opportunity for targeted therapy.” We agree that selection of targeted therapy relies on more than just NGS. However, as demonstrated in table 2, the commercially available NGS platforms used by clinicians often reflexively include immunohistochemical testing for various biomarkers such as PD-L1, MMR, and ER/PR positivity among others. This streamlines the process for practicing clinicians as they only need to order and interpret one diagnostic test when determining the best therapy. We have also attempted to clarify throughout the manuscript which tests are obtained via IHC vs NGS.
Third, immunotherapy in ovarian cancer has not demonstrated a role to date. The authors should include a part on the main molecular pathways that drive ovarian cancer carcinogenesis and the specific genetic alteration involved in the two type pathways (high grade and low grade) instead to devote such a large part of the manuscript to immunotherapy
Author response: Although there are few published trials evaluating immunotherapy exclusively in ovarian cancer, there are several immunotherapy options with tissue agnostic indications that demonstrated equivalent ORR in ovarian cancer patients (Marabelle et al. 2020 PMID 32919526 and Marabelle et al. 2020 31682550). There are also several ongoing clinical trials involving the use of immunotherapy in ovarian cancer. We believe that understanding the mechanism of action and current role of immunotherapy is important for clinicians in anticipation of the results from the current clinical trials.
We agree that the molecular pathways driving carcinogenesis of type 1 and type 2 ovarian cancers are critical for the selection of targeted therapies and chemotherapeutics. However, we believe an in-depth discussion of the molecular and cellular signaling pathways leading to carcinogenesis is outside the scope of this review. We wrote this article to be accessible for the practicing clinician, looking to better understand NGS, molecular biomarkers and targeted therapies. Our references do include several papers and textbooks that provide a more comprehensive discussion of the molecular pathogenesis of ovarian cancers (Lisio et al 2019).
The paragraph 3 should be moved before after the introduction
Author response: Thank you for this insightful recommendation. We believe the reviewer was referencing section 3 of the manuscript (the section covering commercially available NGS platforms). We have moved section 3 to follow the introduction.
The clinical significance of detect actionable mutations and the incidence of these mutations in ovarian cancer should be discussed. This is fundamental to understand the role of NGS in ovarian cancer, that is not limited only to high grade ovarian cancer and to detect BRCA and HRD mutations.
Author response: Thank you for this recommendation. We have included a new table (table 1) that details both the molecular alterations and therapeutic targets based on histologic subtype.
The review is not acceptable in the present form. It should be completely revised and rewritten
Author response: Thank you for this comment. We have updated our manuscript to reflect your recommendations.